# EMERGENT COMMUNICATION THROUGH NEGOTIATION

**Kris Cao**[*]
Department of Computer Science and Technology,
University of Cambridge, UK

**Angeliki Lazaridou, Marc Lanctot, Joel Z Leibo, Karl Tuyls, Stephen Clark**
DeepMind,
London, UK

## ABSTRACT

Multi-agent reinforcement learning offers a way to study how communication could emerge in communities of agents needing to solve specific problems. In this paper, we study the emergence of communication in the negotiation environment, a semi-cooperative model of agent interaction. We introduce two communication protocols – one grounded in the semantics of the game, and one which is *a priori* ungrounded and is a form of cheap talk. We show that self-interested agents can use the pre-grounded communication channel to negotiate fairly, but are unable to effectively use the ungrounded channel. However, prosocial agents do learn to use cheap talk to find an optimal negotiating strategy, suggesting that cooperation is necessary for language to emerge. We also study communication behaviour in a setting where one agent interacts with agents in a community with different levels of prosociality and show how agent identifiability can aid negotiation.

## 1 INTRODUCTION

How can communication emerge? A necessary prerequisite is a task that requires coordination between multiple agents to solve, and some communication protocol for the agents to exchange messages through (see a review by Wagner et al. (2003) on earlier work on emergent communication as well as recent deep reinforcement learning methods by Foerster et al. (2016) and Sukhbaatar et al. (2016)). Given these basic requirements, an interesting question to ask is what task structures aid the emergence of communication and how different communication protocols affect task success.

In the context of linguistic communication, previous work on this subject has mainly studied the emergence of communication in co-operative games like referential games, variants of the Lewis signaling game (Lewis, 1969), where messages are used to disambiguate between different possible referents (Goldman et al., 2007; Lazaridou et al., 2016; Evtimova et al., 2017). Human language, though, is not merely a referential tool. Amongst other things, we communicate private information and thoughts, discuss plans, ask questions and tell jokes. Moreover, many human interactions are not fully cooperative, yet we can still successfully use language to communicate in these situations.

In this paper, we study communication in the negotiation game (see Figure 1), an established model of non-cooperative games in classical game theory (Nash, 1950b; Neumann & Morgenstern, 1944; Nash, 1950a; 1951; Schelling, 1960; Binmore et al., 1986; Peters, 2008). In this game, agents are asked to establish a mutually acceptable division of a common pool of items while having their own hidden utilities for each of them. Effective communication is crucial in this game, as the agents need to exchange strategic information about their desires, infer their opponent's desires from communication, and balance between the two.

Work in classical game theory on negotiation typically uses simple forms of offer / counter-offer bargaining games that do not explicitly address the question of emergent communication (Rubin-

---

[*]Work done during an internship at DeepMind. Correspondence to Kris Cao (`kc391@cam.ac.uk`) or Angeliki Lazaridou (`angeliki@google.com`).

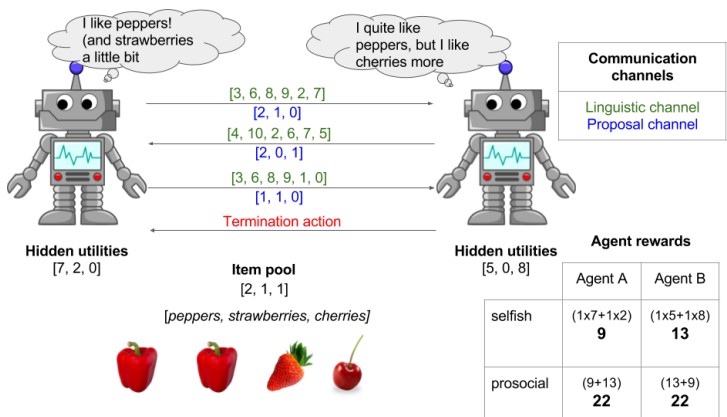

Figure 1: High-level overview of the negotiation environment that we implement. Agent A consistently refers to the agent who goes first.

stein, 1982). Recent work on deep multi-agent reinforcement learning (MARL) has shown great success in teaching agents complex behaviour without a complex environment simulator or demonstration data (Panait & Luke, 2005; Busoniu et al., 2008; Tuyls & Weiss, 2012; Silver et al., 2017; Leibo et al., 2017). By repeatedly interacting with other agents learning at the same time, agents can gradually bootstrap complex behaviour, including motor skills (Bansal et al., 2017) and linguistic communication (Lazaridou et al., 2016; Havrylov & Titov, 2017).

We apply techniques from the MARL literature and train agents to negotiate using task success as the only supervision signal.[1] We show that, when communicating via a task-specific communication channel with inherent semantics, selfish agents can learn to negotiate fairly and divide up the item pool to the agents' mutual satisfaction. However, when communicating via cheap talk (Crawford & Sobel, 1982; Farrell & Rabin, 1996), a task-independent communication channel consisting of sequences of arbitrary symbols similar to language, selfish agents fail to exhibit negotiating behaviour at all. On the other hand, we show that cheap talk can facilitate effective negotiation in prosocial agents, which take the other agent's reward into consideration, providing experimental evidence that cooperation is necessary for language emergence (Nowak & Krakauer, 1999).

The above results are obtained from paired agents interacting exclusively with each other. In more realistic multi-agent scenarios, agents may interact with many other agents within a society. In these cases, cheap talk can have a significant effect on the evolutionary dynamics of the population (Robson, 1990) as well as the equilibria, stability, and basins of attractions (Skyrms, 2002). Furthermore, it is well-known that, unless trained in a diverse environment, agents overfit to the their specific opponent or teammate (Lanctot et al., 2017). Inspired by these considerations, we perform experiments where agents interact with many agents having different prosociality levels, and find that being able to identify and model other agents' beliefs aids the negotiation success. This is consistent with experiments using models based on Theory of Mind: boundedly rational agents can collectively benefit by making inferences about the sophistication levels and beliefs of their opponents, and there is evidence that this occurs in human behavior (Yoshida et al., 2008).

## 2 GAME SETTING

### 2.1 NEGOTIATION ENVIRONMENT

The negotiation task is based on the set-up of Lewis et al. (2017), itself based on DeVault et al. (2015). Agents are presented with three types of items: *peppers*, *cherries* and *strawberries*. At each round (i) an item pool is sampled uniformly, instantiating a quantity (between 0 and 5) for

---

[1]Moody & Saffell (2001) also applied reinforcement learning with non-linear function approximators to economic problems. They trained a recurrent agent to trade securities on the open market using policy gradient methods, similarly to us. However, our agents must create the market from scratch - as there are no established buyers and sellers, the agents participating in the exchange must learn how to make optimal bids.

each of the types and represented as a vector $i \in \{0...5\}^3$ and (ii) each agent $j$ receives a utility function sampled uniformly, which specifies how rewarding one unit of each item is (with item rewards between 0 and 10, and with the constraint that there is at least one item with non-zero utility), represented as a vector $u_j \in \{0...10\}^3$. These utilities are *hidden*, i.e., each agent only has access to its own utilities. Following these assignments, the agents start "negotiating" for $N$ timesteps by exchanging messages $m_t^j$ and proposals $p_t^j \in \{0...5\}^3$ for each timestep $1 \leq t \leq N$. Agents alternate between timesteps, meaning that agent A always acts on the odd turns while agent B on the even turns.

Either agent can terminate the negotiation at any timestep $t$ with a special action, which signifies agreement with the most recent proposal made by the other agent. For example, if agent B terminates the negotiation at $t = 4$, agent A gets rewarded by $R_A = u_A \cdot (p_{t-1}^A)$ while agent B receives reward $R_B = u_B \cdot (i - p_{t-1}^A)$, where $\cdot$ denotes the dot product of vectors. Note that, if an agent makes an invalid proposal, such as asking for more items than exist in the item pool, and this is accepted, then both agents receive $R_A = R_B = 0$.

We impose an upper limit, $N$, on the number of negotiation turns allowed. If the agents get to the limit without agreement, then both agents receive no reward. We experimented with having a fixed upper limit of 10 turns. However, this led to a "first-mover" advantage: if the agents know in advance how many turns the negotiation lasts, there is a degenerate policy in which agent A can wait until the last timestep $N - 1$ before giving a lopsided offer, which agent B is compelled to accept, a situation similar to the ultimatum game (Güth et al., 1982). To eliminate this effect, we sample N between 4 and 10 at each round, according to a truncated Poisson distribution with mean 7.

## 2.2 COMMUNICATION CHANNELS

To achieve negotiation, the agents need to communicate. One obvious communication protocol is to directly transmit the proposed division of items $p_j$. We refer to this as the *proposal* channel. This communication channel is task-specific and pertains to the action space of the agents used to derive their pay-offs. Consequently, the information bandwidth is restricted and grounded in the action space of the game. Motivated by work on emergent communication, we also give our agents the ability of transmitting strings of arbitrary symbols with no *a priori* grounding. We call this the *linguistic* channel, a specific instantiation of the more general concept of cheap talk. This channel differs from the proposal channel in two key properties:

- Non-bindingness: Messages sent via this channel do not commit the sender to any course of action, unlike directly transmitting a proposal which binds the sender to the proposal.

- Unverifiability: There is no inherent link between the linguistic utterance and the private proposal made, meaning that the agents could potentially lie.

We are interested in whether and under what circumstances the agents can make use of this channel to facilitate negotiation and establish a common ground for the symbols.

In our experiments, we consider 4 separate communication configurations: only the proposal channel, only the linguistic channel, both channels open, and no communication at all.[2] If a channel is closed, we replace any messages in that channel with a fixed dummy symbol to ensure no information is transmitted. If the proposal channel is closed, we still use the decoded proposal to calculate the item division, but this information is *not* revealed to the opponent agent.

## 2.3 AGENT SOCIALITY AND REWARD SCHEMES

While purely self-interested agents may learn to divide up items so that only each individual is satisfied, this may not lead to an optimal joint allocation of items, where each item goes to the agent with the highest utility for that item. We introduce a "prosocial" reward $R$ that is shared across A and B, which is the sum of the agents' individual (selfish) rewards, i.e., $R = R_A + R_B$. Prosocial agents are incentivised to communicate, as finding the optimal joint allocation requires

---

[2]We include the no communication baseline to test whether the agents learn to improve over simple heuristics, particularly in the case of two prosocial agents, where a simple strategy such as one agent taking all items with non-zero utility may do unexpectedly well.

communicating the hidden values between the agents. These different reward schemes can be seen as specific instantiations of a more general reward formula of the form $R = \alpha R_A + \beta R_B$ (Peysakhovich & Lerer, 2017). For the selfish agent, $\alpha = 1$ and $\beta = 0$, while for the prosocial agent $\alpha = \beta = 1$.[3]

## 2.4 Agent architecture and learning

At each timestep $t$, the proposer receives three inputs:

- The item context $c^j = [i; u_j]$, a concatenation of the item pool and the proposer's utilities.
- The utterance $m_{t-1}$ produced by the other agent in the previous timestep $t - 1$. If the linguistic channel is closed, this is simply a dummy message.
- The proposal $p_{t-1}$ made by the other agent in the previous timestep $t - 1$. If the proposal channel is closed, this is again a dummy proposal.

First, the discrete inputs are turned into dense vectors through an embedding table. We use two embedding tables, one for the item context and the previous proposal, and a separate one for the previous utterance. This is essential as the item context and the previous proposal have predefined *numeric* semantics, distinct from the *linguistic* semantics of the utterances' symbols.

Then, each of the input sequences is encoded using an LSTM (Hochreiter & Schmidhuber, 1997), one for each input. This results in 3 fixed-size vectors: $h_t^c$, $h_t^m$ and $h_t^p$. These vectors are concatenated and fed through a feedforward layer, followed by a ReLU non-linearity (Nair & Hinton, 2010), to give $h_t$, the hidden state of the agent at timestep $t$. This hidden state is then used to initialise the policies for the actions the agent can take:

- $\pi_{term}$ is the policy for the termination action. If this action is taken by an agent, both agents receive reward according to the last proposal made by the other agent. This is a binary decision, and we parametrise $\pi_{term}$ as a single feedforward layer, with the hidden state as input, followed by a sigmoid function, to represent the probability of termination.
- $\pi_{utt}$ is the policy for the linguistic utterances. This is parametrised by an LSTM, which takes the hidden state of the agent as the initial hidden state. For the first timestep, a dummy symbol is fed in as input; subsequently, the model prediction from the previous timestep is fed in as input at the next timestep, in order to predict the next symbol.
- $\pi_{prop}$ is the policy for the proposals the agent generates. This is parametrised by 3 separate feedforward neural networks, one for each item type, which each take as input $h_t$ and output a distribution over $\{0...5\}$ indicating the proposal for that item.

The overall policy for the agent, $\pi^*$, is a combination of the separate policies. The action that the agent takes at turn $t$ can be summarised by a triple $(e_t, m_t, p_t)$, where $e_t$ is a binary variable indicating whether the agent took the termination action, $m_t$ is a sequence of symbols produced from $\pi_{utt}$, and $p_t$ is a proposal produced by $\pi_{prop}$. A negotiation can then be thought of as a sequence of triples $\tau = \langle (e_t, u_t, p_t) | 1 \leq t \leq N \rangle$, with the agents alternating turns.

During training, each agent $i$ tries to independently find the policy $\pi_i^*$ that maximises the following objective function:

$$\pi_i^* = \arg\max_{\pi_i} \mathbb{E}_{\tau \sim (\pi_A, \pi_B)} [R_i(\tau)] + \lambda H(\pi_i) \qquad (1)$$

where $R_i(\tau)$ is the reward agent $i$ receives from the trajectory $\tau$ and $H(\pi_i)$ is an entropy regularisation term to encourage the agent to explore during training. Note that each agent has its own objective function, and the objectives of both agents are coupled by the trajectory sampled from both agents' policies. At test time, instead of sampling from the policy $\pi$, we take the action with the highest probability. Parameters are updated using the REINFORCE (Williams, 1992) update rule with an exponentially smoothed mean baseline. Full hyperparameter details are in the appendix.

---

[3]To make learning more stable and to make results across different game instantiations more comparable, we also scale the reward obtained between 0 and 1. For the selfish reward scheme, the scaling factor is the total reward achievable for each agent. For the prosocial reward, this is the dot product between the item pool and the element-wise maximum of both agents' utilities. This scaled value is the score reported in the results below.

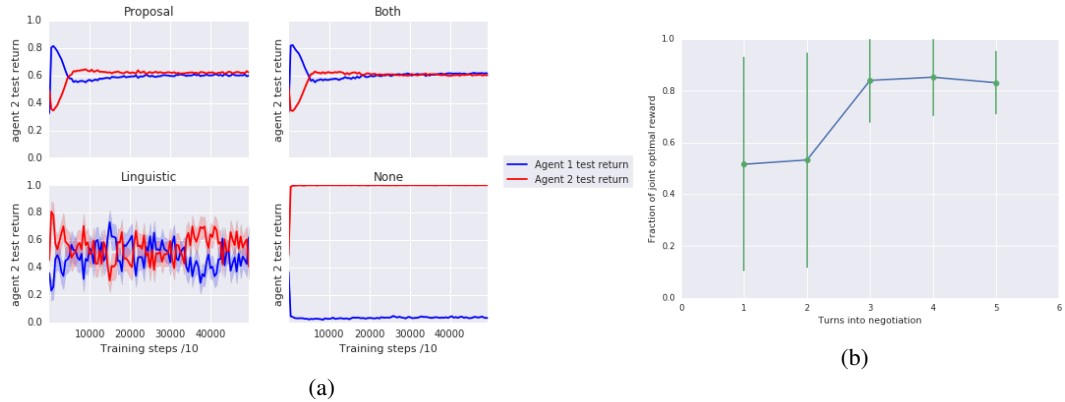

Figure 2: a) Training curves for self-interested agents learning to negotiate under the various communication channels. The results show the mean across 20 different random seeds, as well as bootstrapped confidence intervals via shading (only visible for the linguistic communication case)
b) The optimality of the proposed item division as negotiation proceeds for two selfish agents communicating via the proposal channel, shown with error bars for interquartile range.

# 3 EXPERIMENT 1: CAN SELF-INTERESTED AGENTS LEARN TO NEGOTIATE?

## 3.1 EXPERIMENT DESCRIPTION

In our first experiment, we test whether purely self-interested agents can learn to negotiate and divide up items fairly, and investigate the effect of the various communication channels on negotiation success. We train self-interested agents to negotiate for 500k episodes. Each episode corresponds to a batch of 128 games, each with item pools and hidden utilities generated as described in Section 2.1. We also hold out 5 batches worth of games as a test set, and test the agents every 50 episodes, in order to measure the robustness of the negotiating behaviour to unseen environments.

## 3.2 RESULTS

**Self-interested agents can learn to negotiate fairly**  Figure 2a shows that self-interested agents learn to divide up the items fairly when they exchange proposals directly. We see that the proportion of total utility each agent receives is roughly equal, and above 50%, suggesting that the agents have learnt to keep items with higher utility, while giving away items with low utility.[4] In addition, the agents seem to show evidence of compromise: there is a drop of 0.24 between the value of the proposal that agent 1 makes at the beginning of negotiation, and the final reward it receives; for agent 2, the corresponding drop is 0.18. Furthermore, Figure 2b shows the mean optimality of the proposal made at each turn of the negotiation averaged over 1280 games, calculated as the sum of the reward of both agents divided by the maximum possible reward. This value increases with more rounds of negotiation, suggesting that the agents are adjusting their proposals based on information from the other agent.

Table 1 presents an illustrative example of two self-interested agents negotiating using the proposal channel. While the initial proposals made by both agents are over-optimistic, over the course of the negotiation, they compromise to less items. This allows them to achieve higher joint reward, as seen in Figure 2b, a clear example of "negotiation".

**Self-interested agents do not appear to ground cheap talk**  When using the linguistic channel, agents do not negotiate optimally. Instead, the agents randomly alternate taking all the items, which is borne out by the oscillations in Figure 2a. Indeed, examination of the messages exchanged by self-interested agents show they predominantly consist of a single symbol – see Figures 5a and 5b in the appendix for more details. This result suggests that the self-interested agents we implement

---

[4]Lewis et al. (2017), who crowd-sourced human negotiation data, report that the average human reward is also 60% of the total available reward when negotiating in a similar environment.

| Item pool: [5, 5, 1] | A utilities: [8, 7, 1] | B utilities: [8, 3, 2] |
|---|---|---|
| Turn | Agent | Proposal |
| 1 | A | [3, 4, 4] |
| 2 | B | [4, 2, 0] |
| 3 | A | [3, 4, 0] |
| 4 | B | [4, 1, 0] |

Table 1: Actual transcript of two self-interested agents negotiating using the proposal channel. In this setting, no linguistic message is decoded at all.

| Agent sociality | | Proposal | Linguistic | Both | None |
|---|---|---|---|---|---|
| Self-interested | Fraction of joint reward | $0.87 \pm 0.11$ | $0.75 \pm 0.22$ | $0.87 \pm 0.12$ | $0.77 \pm 0.22$ |
| | $25^{th}$ & $75^{th}$ percentiles | [0.81, 0.95] | [0.61, 0.94] | [0.81, 0.95] | [0.62, 0.97] |
| | Turns taken | $3.55 \pm 1.12$ | $5.36 \pm 1.20$ | $3.43 \pm 1.10$ | $3.00 \pm 0.13$ |
| Prosocial | Fraction of joint reward | $0.93 \pm 0.10$ | $0.99 \pm 0.02$ | $0.92 \pm 0.11$ | $0.95 \pm 0.11$ |
| | $25^{th}$ & $75^{th}$ percentiles | [0.89, 1.0] | [1.0, 1.0] | [0.88, 1.0] | [0.93, 1.0] |
| | Turns taken | $3.10 \pm 0.99$ | $3.71 \pm 0.58$ | $2.98 \pm 0.97$ | $2.27 \pm 0.69$ |

Table 2: Joint reward success and average number of turns taken for paired agents negotiating with random game termination, varying the agent reward scheme and communication channel. The results are averaged across 20 seeds, with 128 games per seed. We also report the standard deviation as the $\pm$ number and the interquartile range.

find it difficult to cheap talk to exchange meaningful information about how to divide up the items, which is in line with the theoretical analysis of cheap talk in Crawford & Sobel (1982), who find that, once agent interests diverge by a finite amount, no communication is to be expected. Our findings also corroborate the hypothesis found in the language evolution literature that cooperation is a prerequisite for the emergence of language (Nowak & Krakauer, 1999).

We conjecture that this finding is partially due to the fact that the agents operate in a non-iterated environment (i.e., they have no explicit memory of previous interactions). In the classic prisoner's dilemma game, rational agents playing a single-shot version of the game learn to always defect, whereas rational agents playing iterated prisoner's dilemma can learn more mutually beneficial cooperative strategies under the right conditions (Sandholm & Crites, 1996; Wunder et al., 2010). More sophisticated players will often have an incentive to take a short-term loss in reward to teach or punish another player for long-term gain in the repeated setting (Fudenberg & Levine, 1998). We suspect that, by iterating the game, there is an incentive for the cheap talk to be more verifiable, as there is a more credible threat of punishment if the accepted proposal and the linguistic utterance do not correspond.

## 4 EXPERIMENT 2 - CAN PROSOCIAL AGENTS LEARN TO COORDINATE?

### 4.1 EXPERIMENT DESCRIPTION

In the previous experiment, we showed that self-interested agents can divide up items fairly such that each agent gets equal reward. However, this does not correspond to an optimal joint allocation where items go to the agent with the higher utility for that item (equivalently, maximising the sum of both agents' rewards). The prosocial reward scheme aligns exactly with this optimal outcome, as learning to negotiate equates to finding the optimal joint item allocation. In this setting, there is an in-built bias towards communication, as solving the game optimally requires pooling hidden utilities. Table 2 reports the turns taken as well as the *joint reward optimality*, i.e., the sum of both agents' rewards, divided by the maximum possible reward given the item context, where the latter corresponds to each agent getting all the items for which they have a higher utility.

| Item pool: [4, 5, 1] | A utilities: [1, 7, 9] | | B utilities: [7, 0, 10] |
|---|---|---|---|
| Turn | Agent | Linguistic utterance | Proposal (hidden) |
| 1 | A | [3, 3, 3, 3, 3, 3] | [0, 5, 1] |
| 2 | B | [6, 6, 6, 4, 7, 4] | [4, 0, 1] |
| 3 | A | [3, 3, 3, 3, 3, 3] | [0, 5, 0] |

Table 3: Actual transcript of two prosocial agents negotiating using the linguistic channel. At each turn, the agents decode a proposal but this is hidden and not revealed to the other agent.

## 4.2 RESULTS

**Cheap talk helps agents coordinate**   For prosocial agents with aligned interests, theoretical results suggest that communication using cheap talk is a Nash equilibrium. This is indeed what we observe, as the linguistic channel results in much better task success than any other communication scheme. The information bandwidth of the proposal channel is limited by the type and number of items, whereas the linguistic channel's bandwidth is unconstrained and can (in theory) be arbitrarily large, helping the effective exchange of task-specific information. In addition, the variance in joint optimality is much lower for the linguistic channel, suggesting it is more robust. We suspect that difficulties in optimisation cause the results for both channels open to be similar to those for only the proposal channel open, as the proposal channel is pre-grounded, which poses a strong local maxima. With better optimisation, we believe that the performance of both channels open will more closely match those of just the linguistic channel open.

Table 3 illustrates an example of negotiation between the two prosocial agents. Interestingly, we observe that agent A over the course of the negotiation game adjusts its proposal (which is however hidden and not revealed) based on the utterances of agent B.

**Prosocial agents can still use the proposal channel to co-ordinate**   With random termination, agents communicating with the proposal channel do even worse than the no-communication baseline. However, when given the full 10 turns to exchange information (see Figure 7 in the appendix), prosocial agents manage to outperform the no-communication baseline. This suggests that the agents are slowly learning to repurpose the proposal channel to transmit information, i.e., they communicate directly on the action space. Reusing task-specific actions to transmit information has precedent, such as in bridge bidding (Simon, 1949), and it is interesting to observe that agents learn to develop a codebook without the need for prior agreement.

## 5 ANALYSIS OF LINGUISTIC COMMUNICATION

### 5.1 SYMBOL USAGE

The symbol unigram and bigram distributions of messages exchanged by prosocial agents (see Figures 3a and 3b respectively) show that agent A, the agent who initiates the negotiation, is not transmitting any information using the linguistic channel. On the other hand, agent B uses a diversity of symbols resulting in a long-tailed bigram symbol distribution, reminiscent of the Zipfian distribution found in natural languages. This suggests that, even though the task is symmetric, the agents differentiate into speaker and listener roles, similar to other co-ordination games of communication, such as the reference game. Thus, they have adopted a simple strategy for solving the game: an agent shares its utilities (the speaker), while the other (the listener) compares the shared utilities with their own and generates the optimal proposal.

In contrast, the selfish agents do not show evidence of grounded symbol usage. The unigram and bigram statistics show that most messages consist solely of strings of a single fixed symbol, regardless of the item context, and hence no information is being exchanged.

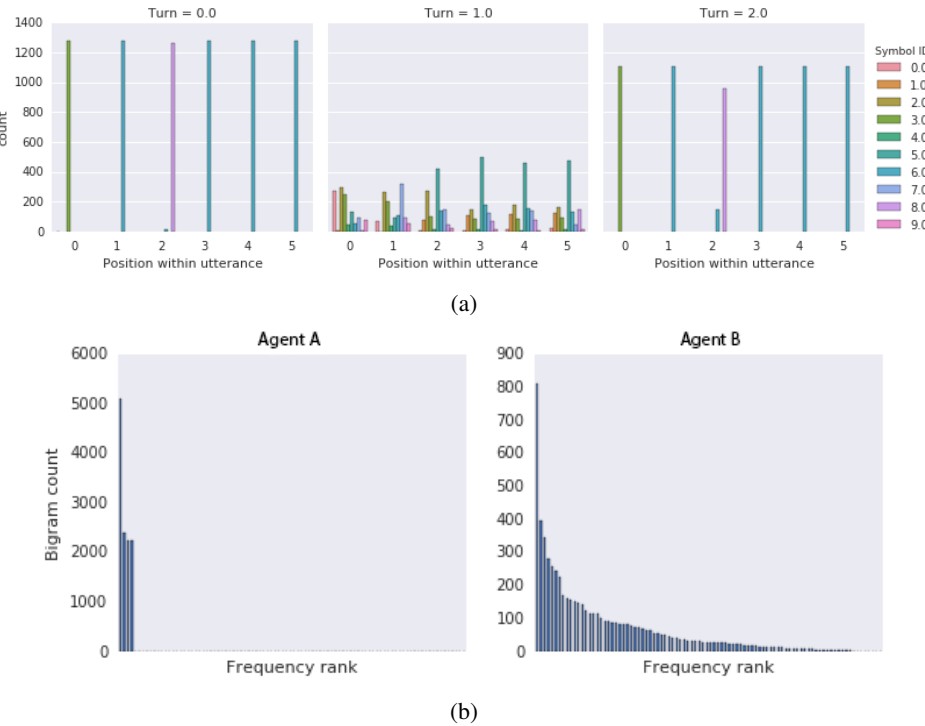

(a)

(b)

Figure 3: a) Unigram statistics of symbol usage broken down by turn and by position within the utterance for prosocial agents communicating via the linguistic channel. b) Bigram counts for prosocial agents communicating via the linguistic channel, sorted by frequency.

| Agent types | % agent A value correct | % agent B value correct | % proposal correct |
|---|---|---|---|
| Selfish-Selfish | 21 | 25 | 94 (94) |
| Prosocial-Prosocial | 26 | 81 | 80 (57) |
| Random baseline | 20 | 20 | 17 |

Table 4: Accuracy of predicting individual elements of the agents' hidden utilities, as well as the final proposal that was accepted. Numbers in brackets indicate accuracy predicting the proposal from just the item pool. Random baseline numbers are obtained by predicting from a message transcript and item pool of all 0's.

## 5.2 CONTENT OF THE MESSAGES

To interpret the information that is transmitted through the linguistic channel, we use the message transcript and the item pool to create probe classifiers that predict the hidden utilities of each agent and the (hidden) accepted proposal. We use an LSTM to encode the sequence of symbols belonging to the message transcript into a vector, and another LSTM to encode the item pool into a vector. The resulting two vectors are then concatenated and used to predict the hidden utility functions and the accepted proposal using 9 linear classifiers, one for each item (3 each for agent A and agent B's utilities and 3 for the proposal). We conduct 10-fold cross-validation and report averaged per-item accuracy (see Table 4).

We also include two baselines. The first, reported as the "random baseline" in Table 4, attempts to predict the accepted proposal and agent utilities from a message transcript and item pool of all 0's (indicating chance performance at the task). The other predicts the accepted proposal from the item pool and a message transcript of all 0's. This shows how much additional information about the proposal is contained in the message transcript.

We can see from Table 4 that not only is there variety in the symbols used, but the messages also have semantic content: they contain information about the hidden utilities of agent B, and about the

proposal that was made. This shows that our agents have learned to give meaning to the symbols, and can use them to transmit information. However, the purely self-interested agents do not seem to transmit meaningful information using the linguistic channel, and thus do not appear to ground the symbols.[5]

# 6    EXPERIMENT 3: A SOCIETY OF AGENTS

In more realistic scenarios, we learn and practice negotiation in environments with diverse agent populations and levels of prosociality. In this setting, maximising one's reward requires identifying which agents are most useful in achieving one's aims. In practice, this involves identifying which agents are the most prosocial, as they are both the most exploitable for self-interested agents, and the most cooperative for other prosocial agents.

## 6.1    EXPERIMENTAL PROTOCOL

We trained a fixed agent against a community of 10 agents, with varying proportions of selfish and prosocial agents. For each training episode, we randomly sample one agent from the community, play a batch of negotiation games, and then update both agents. We do this for both fixed agent A (the agent who goes first) and for fixed agent B (the agent who goes second). We experiment with either 1 or 5 prosocial agents, which communication channels are open, and also whether the agents are identifiable.

At test time, we care primarily about whether self-interested agents can exploit prosocial agents and whether prosocial agents can cooperate with other prosocial agents. Hence, we only test the fixed agent against the prosocial agents in the community. We generate 10 batches of 128 games each batch, and play each game with every pair of fixed agent and community prosocial agent. We average the results across all pairs and all games, and show the mean and standard deviations of the rewards obtained in this experiment in Table 5.

If opposing agents are identifiable (i.e. each agent has a name-tag), the fixed agent receives this information as a one-hot vector, which is then used to look up an embedding table with trainable parameters with one embedding per opposing agent. This opponent embedding is then concatenated with the encoded inputs, before being passed through the feedforward layer to get the agent's hidden state. Intuitively, the induced agent embedding, that takes the form of machine theory of mind (Rabinowitz et al., 2018), should capture any information about the opponent's behaviour (e.g., negotiating strategy) that, when taken into account, can help an agent maximize their reward.

## 6.2    ANALYSIS

**Identifying agents**    Our results show that providing agent ID information does help the fixed agent achieve its aims. This effect is particularly pronounced for a selfish fixed agent; here, providing ID information uniformly improves performance. For cooperative agents, the results mixed; for a fixed agent A communicating via the linguistic channel, having agent ID information harms performance. Indeed, the only prosocial agents who beat the no communication baseline (which achieved an average joint optimal reward of 0.95) was when agent A was fixed and ID information was not provided.

We also computed the 2D PCA projections of the learnt opponent embeddings for a fixed agent A under a variety of channels and prosocial levels (see Figure 4). Even in cases where the agent ID does not aid negotiation, we find that the embeddings cluster according to the reward scheme of the other agent. This shows that the agents can distinguish different agents by their reward schemes from purely observing negotiating behaviour.

**Community linguistic phenomena**    In one of our experimental settings, a community of prosocial agents developed a language and were able to use this to achieve better negotiation success. This is the outcome in the starred cell in Table 5. For a visualisation of the bigram usage statistics in

---

[5]The high proposal accuracy is because the agents propose taking all the items on each turn, regardless of the messages exchanged, which correlates highly with the item pool itself.

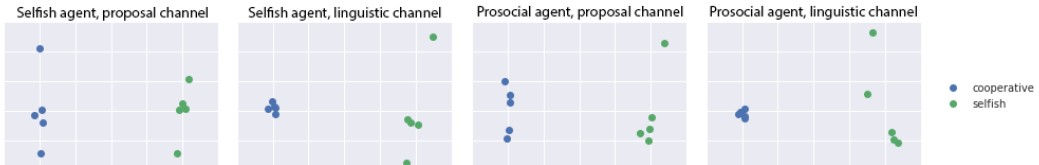

Figure 4: PCA plot of whitened opponent ID embeddings that was learnt by a fixed agent 1 for a variety of reward schemes and communication channels.

| Fixed agent type | # proso-cial agents | IDs given | Fixed agent A | | | Fixed agent B | | |
|---|---|---|---|---|---|---|---|---|
| | | | Proposal | Linguistic | Both | Proposal | Linguistic | Both |
| Self | 1 | False | $0.81 \pm 0.17$ | $\mathbf{0.97 \pm 0.05}$ | $0.87 \pm 0.11$ | $0.76 \pm 0.18$ | $\mathbf{1.0 \pm 0.0}$ | $0.76 \pm 0.16$ |
| | | True | $\mathbf{0.96 \pm 0.06}$ | $\mathbf{0.97 \pm 0.10}$ | $\mathbf{0.97 \pm 0.04}$ | $\mathbf{0.95 \pm 0.08}$ | $\mathbf{1.0 \pm 0.0}$ | $\mathbf{1.0 \pm 0.0}$ |
| | 5 | False | $0.82 \pm 0.17$ | $\mathbf{1.0 \pm 0.0}$ | $0.98 \pm 0.07$ | $0.88 \pm 0.14$ | $\mathbf{1.0 \pm 0.0}$ | $0.88 \pm 0.14$ |
| | | True | $\mathbf{1.0 \pm 0.01}$ | $\mathbf{1.0 \pm 0.0}$ | $\mathbf{1.0 \pm 0.01}$ | $\mathbf{1.0 \pm 0.0}$ | $\mathbf{1.0 \pm 0.0}$ | $\mathbf{1.0 \pm 0.02}$ |
| Pros. | 1 | False | $0.91 \pm 0.11$ | $0.95 \pm 0.09$ | $0.91 \pm 0.12$ | $0.93 \pm 0.11$ | $0.95 \pm 0.10$ | $0.92 \pm 0.11$ |
| | | True | $0.95 \pm 0.10$ | $0.93 \pm 0.11$ | $0.95 \pm 0.09$ | $0.92 \pm 0.01$ | $0.96 \pm 0.08$ | $0.92 \pm 0.10$ |
| | 5 | False | $0.90 \pm 0.13$ | $0.98^* \pm 0.06$ | $0.93 \pm 0.10$ | $0.92 \pm 0.12$ | $0.95 \pm 0.08$ | $0.90 \pm 0.16$ |
| | | True | $0.94 \pm 0.09$ | $0.97 \pm 0.06$ | $0.94 \pm 0.01$ | $0.93 \pm 0.10$ | $0.95 \pm 0.08$ | $0.92 \pm 0.10$ |

Table 5: Fixed agent return vs opposing prosocial agents, for different numbers of opposing prosocial agents and whether IDs are revealed across the communication channels. For the self-interested fixed agent, the numbers report personal reward, and for the prosocial fixed agent, the numbers report joint reward. The starred result indicates when a community has made use of the linguistic channel to transmit information.

each interaction pair in this community, see Figure 6. Interestingly, the only situation when agents developed a language was when ID information was not provided. We suspect this result is due to a poor local optimum; since the prosocial agents can learn to use IDs to distinguish prosocial agents from the selfish agents they can ignore the linguistic utterance.

Moreover, we find that when prosocial agents make use of the linguistic channel, the communication protocol differs within the community. We calculate the Spearman correlation $\rho$ between bigram ranks of different pairs of agents, all of them with the same fixed agent A, and show the results in Table 6. Even though all pairs of agents share the same fixed agent A, different pairs of agents learn to use bigrams in a different way, resulting in relatively low correlations between -0.22 and 0.27.

# 7 DISCUSSION

We showed that by communicating through a verifiable and binding communication channel, self-interested agents can learn to negotiate fairly by reinforcement learning, using only task success as the reward signal. Moreover, cheap talk facilitated negotiation in prosocial but not in self-interested agents, corroborating theoretical results from the game theory literature (Crawford & Sobel, 1982). An interesting future direction of research would be to investigate whether cheap talk can be made to emerge out of self-interested agents interacting. Recent encouraging results by Crandall et al. (2018) show that communication can help agents cooperate. However, their signalling mechanism is heavily engineered: the speech acts are predefined and the consequences of the speech acts on the observed behaviour are deterministically hard-coded. It would be interesting to see whether a learning algorithm, such as Foerster et al. (2017), can discover the same result.

A related paper from Lewis et al. (2017) takes a top-down approach to learning to negotiate by leveraging dialogue data. We demonstrated a bottom up alternative towards learning communicative behaviours directly from interaction with peers. This opens up the exciting possibility of learning

domain-specific reasoning capabilities from interaction, while having a general-purpose language layer at the top producing natural language.

## ACKNOWLEDGEMENTS

We would like to thank Mike Johanson for his insightful comments on an earlier version of this paper, as well as Karl Moritz Hermann and the rest of the DeepMind language team for many fruitful discussions over the course of the project.

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

## A    ADDITIONAL FIGURES AND TABLES

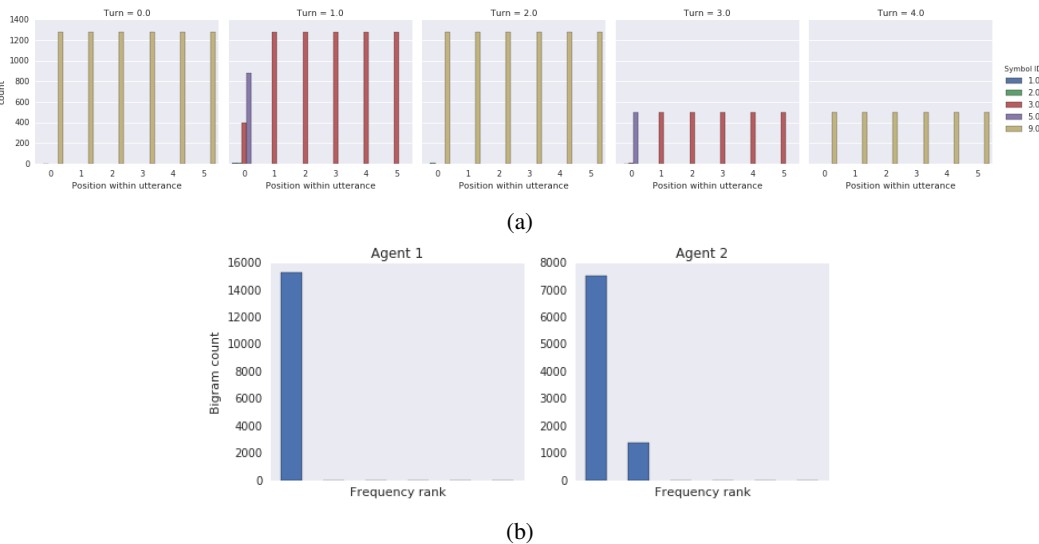

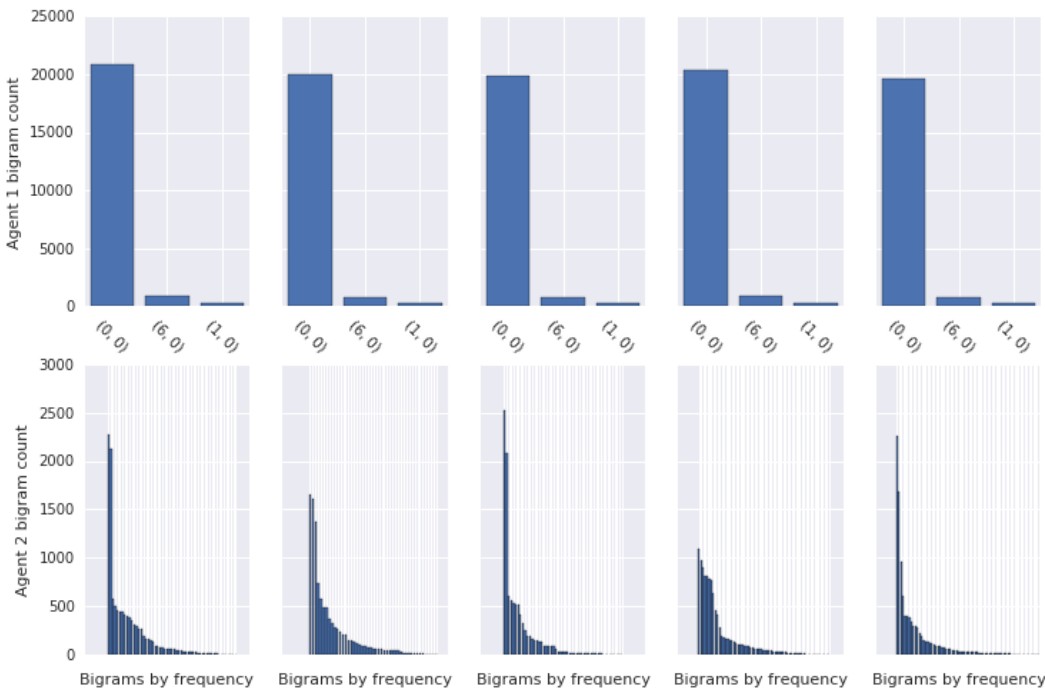

Figure 5: a) Unigram statistics of symbol usage broken down by turn and by position within the utterance for selfish agents communicating via the linguistic channel. b) Bigram counts for selfish agents communicating via the linguistic channel, sorted by frequency.

Figure 6: Bigram usage in all interaction pairs between a fixed prosocial agent A and a community of 5 prosocial agent Bs. This is the only case where the average joint optimality reward of the negotiating agents is higher than the no communication baseline.

|            | Agent 2 ID 1 | Agent 2 ID 2 | Agent 2 ID 3 | Agent 2 ID 4 |
|------------|--------------|--------------|--------------|--------------|
| Agent 2 ID 0 | -0.118     | 0.132        | 0.101        | 0.057        |
| Agent 2 ID 1 |            | -0.019       | -0.228       | 0.269        |
| Agent 2 ID 2 |            |              | 0.178        | -0.165       |
| Agent 2 ID 3 |            |              |              | -0.067       |

Table 6: Correlation between bigram usage between the different responders in the community.

| Agent sociality | | Proposal | Linguistic | Both | None |
|-----------------|---|----------|------------|------|------|
| Self-interested | Fraction of joint reward | $0.76 \pm 0.27$ | $0.74 \pm 0.23$ | $0.76 \pm 0.28$ | $0.23 \pm 0.06$ |
|                 | $25^{th}$ & $75^{th}$ percentiles | [0.66, 0.96] | [0.59, 0.96] | [0.68, 0.96] | [0.19, 0.29] |
|                 | Turns taken | $9.99 \pm 0.35$ | $4.29 \pm 2.13$ | $10.0 \pm 0.25$ | $7.90 \pm 3.21$ |
| Prosocial       | Fraction of joint reward | $0.96 \pm 0.07$ | $0.95 \pm 0.10$ | $0.96 \pm 0.07$ | $0.85 \pm 0.31$ |
|                 | $25^{th}$ & $75^{th}$ percentiles | [0.95, 1.0] | [0.91, 1.0] | [0.94, 1.0] | [0.90, 1.0] |
|                 | Turns taken | $9.06 \pm 2.50$ | $4.56 \pm 2.83$ | $9.06 \pm 2.41$ | $3.30 \pm 2.50$ |

Table 7: Joint reward success and average number of turns taken for paired agents negotiating when allowed the full 10 turns, varying the agent reward scheme and communication channel. The results are averaged across 20 seeds, with 128 games per seed. We also report the standard deviation as the $\pm$ number and the quartiles.

## B    HYPERPARAMETER DETAILS

Embedding sizes, and all neural network hidden states, had dimension 100. We used the ADAM optimizer (Kingma & Ba, 2014), with default parameter settings, to optimize the parameters of each agent. Each agent had a separate optimizer. We used a separate value of $\lambda$, the entropy regularisation weight hyperparameter, for each policy. For $\pi_{term}$ and $\pi_{prop}$, $\lambda = 0.05$; for $\pi_{utt}$, $\lambda = 0.001$. The symbol vocabulary size was 11, and the agents were allowed to generate utterances of up to length 6. The smoothing constant for the exponential moving average baseline was 0.7 (i.e. if the old baseline value was $b_{old}$, and the current reward is $R$, then the new estimate of the baseline is $b_{new} = 0.7b_{old} + 0.3R$).

