# OpenReview forum: "Emergent Communication through Negotiation"
_ICLR.cc/2018/Conference — Accept (Poster)_

### Official Review · AnonReviewer1 · 2017-11-27
**Review 'Emergent Communication through Negotiation'**

**Rating:** 6
**Confidence:** 3

**Review:**

The authors describe a variant of the negotiation game in which agents of different type, selfish or prosocial, and with different preferences. The central feature is the consideration of a secondary communication (linguistic) channel for the purpose of cheap talk, i.e. talk whose semantics are not laid out a priori.

The essential findings include that prosociality is a prerequisite for effective communication (i.e. formation of meaningful communication on the linguistic channel), and furthermore, that the secondary channel helps improve the negotiation outcomes.

The paper is well-structured and incrementally introduces the added features and includes staged evaluations for the individual additions, starting with the differentiation of agent characteristics, explored with combination of linguistic and proposal channel. Finally, agent societies are represented by injecting individuals' ID into the input representation.

The positive:
- The authors attack the challenging task of given agents a means to develop communication patterns without apriori knowledge.
- The paper presents the problem in a well-structured manner and sufficient clarity to retrace the essential contribution (minor points for improvement).
- The quality of the text is very high and error-free.
- The background and results are well-contextualised with relevant related work.

The problematic:
- By the very nature of the employed learning mechanisms, the provided solution provides little insight into what the emerging communication is really about. In my view, the lack of interpretable semantics hardly warrants a reference to 'cheap talk'. As such the expectations set by the well-developed introduction and background sections are moderated over the course of the paper.
- The goal of providing agents with richer communicative ability without providing prior grounding is challenging, since agents need to learn about communication partners at runtime. But it appears as of the main contribution of the paper can be reduced to the decomposition of the learnable feature space into two communication channels. The implicit relationship of linguistic channel on proposal channel input based on the time information (Page 4, top) provides agents with extended inputs, thus enabling a more nuanced learning based on the relationship of proposal and linguistic channel. As such the well-defined semantics of the proposal channel effectively act as the grounding for the linguistic channel. This, then, could have been equally achieved by providing agents with a richer input structure mediated by a single channel. From this perspective, the solution offers limited surprises. The improvement of accuracy in the context of agent societies based on provided ID follows the same pattern of extending the input features.
- One of the motivating factors of using cheap talk is the exploitation of lying on the part of the agents. However, apart from this initial statement, this feature is not explicitly picked up. In combination with the previous point, the necessity/value of the additional communication channel is unclear.

Concrete suggestions for improvement:

- Providing exemplified communication traces would help the reader appreciate the complexity of the problem addressed by the paper.
- Figure 3 is really hard to read/interpret. The same applies to Figure 4 (although less critical in this case).
- Input parameters could have been made explicit in order to facilitate a more comprehensive understanding of technicalities (e.g. in appendix).
- Emergent communication is effectively unidirectional, with one agent as listener. Have you observed other outcomes in your evaluation?

In summary, the paper presents an interesting approach to combine unsupervised learning with multiple communication channels to improve learning of preferences in a well-established negotiation game. The problem is addressed systematically and well-presented, but can leave the reader with the impression that the secondary channel, apart from decomposing the model, does not provide conceptual benefit over introducing a richer feature space that can be exploited by the learning mechanisms. Combined with the lack of specific cheap talk features, the use of actual cheap talk is rather abstract. Those aspects warrant justification.

---

> ### Author Response · Authors · 2018-01-03
> **Reply to AnonReviewer 1**
>
> We would like to emphasize that the linguistic/utterance channel and the proposal channel are completely separate, and there is no a priori link in the messages in the proposal channel and the messages in the linguistic channel. When the agents are forced to use the linguistic channel exclusively, they must learn from scratch how to use the linguistic channel to communicate effectively to solve the negotiation task and thus the proposal channel is never communicated to the opponent. In short, LINGUISTIC refers to negotiating using ONLY the linguistic channel, PROPOSAL only the preference and BOTH using the combination.
>
> >By the very nature of the employed learning mechanisms, the provided solution provides little insight into what the emerging communication is really about.
>
> It is difficult to quantify precisely what communication is about, especially in our bottom-up approach starting from arbitrary symbols. Despite this, in our post-processing analyses of the communication analyses, we show: (i) agents partition themselves into speaker and listener, (ii) elements of natural language are found in the protocols that emerged, and (iii) the content of the messages indicate that the agents are encoding their utilities in the language channel.
>
> >In my view, the lack of interpretable semantics hardly warrants a reference to 'cheap talk'. As such the expectations set by the well-developed introduction and background sections are moderated over the course of the paper.
>
> We did not intend to suggest that the lack of interpretable semantics warrants a reference to cheap talk. We refer to cheap talk due to the fact that the exchanges in the linguistic channel have no effect on the resulting payoff, which follows directly from the definition.
> The lack of interpretable semantics is orthogonal to any references to cheap talk: it simply motivates the research question of whether communication can emerge among learning agents. We will clarify this.
>
> >Providing exemplified communication traces would help the reader appreciate the complexity of the problem addressed by the paper.
>
> In our most recent revision, we have added an appendix showing what sample games with each of the communication channels open look like.
>
> >Figure 3 is really hard to read/interpret. The same applies to Figure 4 (although less critical in this case).
>
> We have made the figures larger, and added more explanation in the text.
>
> >Input parameters could have been made explicit in order to facilitate a more comprehensive understanding of technicalities (e.g. in appendix).
>
> We have added an appendix showing the values of the hyperparameters we used. We would also like to thank the public comments that acted as additional motivation to help reproducibility.
>
> >Emergent communication is effectively unidirectional, with one agent as listener. Have you observed other outcomes in your evaluation?
>
> In our experiments, we consistently see the agents separating into speaker-listener roles, as mentioned in the paper.

---

> > ### Comment · AnonReviewer1 · 2018-01-12
> > **Thanks for the response.**
> >
> > Thank you for responding to the mentioned concerns and addressing those in your latest revision. The topic is interesting and deserves visibility.

---

### Official Review · AnonReviewer2 · 2017-11-27
**Interesting**

**Rating:** 7
**Confidence:** 4

**Review:**

This paper explores how agents can learn to communicate to solve a negotiation task. They explore several settings: grounded vs. ungrounded communication, and self-interested vs. prosocial agents. The main findings are that prosocial agents are able to learn to ground symbols using RL, but self-interested agents are not. The work is interesting and clearly described, and I think this is an interesting setting for studying emergent communication.

My only major comment is that I’m a bit skeptical about the claim that “self-interested agents cannot ground cheap talk to exchange meaningful information”. Given that the agents’ rewards would be improved if they were able to make agreements, and humans can use ‘cheap talk’ to negotiate, surely the inability to do so here shows a failure of the learning algorithm (rather than a general property of self-interested agents)?

I am also concerned about the dangers posed by robots inventing their own language, perhap the authors should shut this down :-)

---

> ### Author Response · Authors · 2018-01-03
> **Reply to AnonReviewer 2**
>
> >My only major comment is that I’m a bit skeptical about the claim that “self-interested agents cannot ground cheap talk to exchange meaningful information”. Given that the agents’ rewards would be improved if they were able to make agreements, and humans can use ‘cheap talk’ to negotiate, surely the inability to do so here shows a failure of the learning algorithm (rather than a general property of self-interested agents)?
>
> We agree; we believe this is the the main reason that the bottom-up approach is particularly challenging. Humans (and to a lesser extent, the demonstration data in top-down approaches) benefit by having a priori semantics on the symbols in the linguistic channel. We used the term ‘self-interested agents’ mainly to separate them from the prosocial ones, but we do indeed mean in the context of the standard RL learning algorithms used in the paper, not more generally to mean ‘any possible self-interested agent’. We will clarify this.
> In future work, we will explore more sophisticated RL learning techniques that allow self-interested to negotiate using a ‘cheap-talk’ channel (which due to its unbinding and unverifiable nature poses a challenge for the current RL algorithms).

---

### Official Review · AnonReviewer3 · 2017-11-29
**A very interesting paper, it examines problems of how agents can use communication to maximise their rewards in a simple negotiation game. The good results of the paper are muddied by all the other data and aspects of the experiments that didn't work.**

**Rating:** 5
**Confidence:** 4

**Review:**

The experimental setup is clear, although the length of the utterances and the number of symbols in them is not explicitly stated in the text (only the diagrams).

Experiment 1 confirms that agents who seek only to maximise their own rewards fail to coordinate over a non-binding communication channel. The exposition of the experiments, however, is unclear.
In Fig 1, it is not clear what Agent 1 and Agent 2 are. Do they correspond to arbitrary labels or the turns that the agent takes in the game?
Why is Agent 1 the one who triumphs in the no-communication channel game? Is there any advantage to going first generally? Where are the tests of robustness on the curves demonstrated in Figure 2a?
Has figure 2b been cherry picked? This should be demonstrated over many different negotiations with error bars.
In the discussion of the agents being unable to ground cheap talk, the symbolic nature of the linguistic channel clouds the fact that it is not the symbolic, ungrounded aspect but the non-binding nature of communication on this channel. This would be more clearly demonstrated and parsimonious by using a non-binding version of the proposal channel and saving the linguistic discussion for later.

Experiment 2 shows that by making the agents prosocial, they are able to learn to communicate on the linguistic channel to achieve pretty much optimal rewards, a very nice result.
The agents are not able to reach the same levels of cooperation on the proposal channel, in fact performing worse than the no-communication baseline. Protocols could be designed that would allow the agents to communicate their utilities over this channel (within 4 turns), so the fact they don't suggests it is the learning procedure that is not able to find this optimum. Presenting this as a result about the superiority of communication over the linguistic channel is not well supported.
Why do they do worse with random termination than 10 turns in the proposal channel? 4 proposals should contain enough information to determine the utilities.
Why are the 10 turn games even included in this table? It seems that this was dismissed in the environment setup section, due to the first mover advantage.
Why do no-communication baselines change so much between random termination and 10 turns in the prosocial case?
Why do self-interested agents for 10 turns on the linguistic channel terminate early?
Table 1 might be better represented using the median and quartiles, since the data is skewed.

Analysis of the communication, i.e. what is actually sent, is interesting and the division into speaker and listener suggests that this is a simple protocol that is easy for agents to learn.

Experiment 3 aims to determine whether an agent is able to negotiate against a community of other agents with mixed levels of prosociality. It is shown that if the fixed agent is able to identify who they are playing against they can do better than not knowing, in the case where the fixed agent is self interested.
The pca plot of agent id embeddings related is good.
Both Figure 4 and Table 3 use Agent 1 and Agent 2 rather than Agent A and Agent B and is not clear whether this is a mistake or Agent 1 is different from Agent A.
The no-communication baseline is referred to in the text but the results are not shown in the table.
There are no estimates of the uncertainty of the results in table 3, how robust are these results to different initial conditions?
This section seems like a bit of an add-on to address criticisms that might arise about the initial experiment being only two agents.

Overall, the paper has some nice results and an interesting ideas but could do with some tightening up of the results to make it really good.

---

> ### Author Response · Authors · 2018-01-03
> **Reply to AnonReviewer 3**
>
> >In Fig 1, it is not clear what Agent 1 and Agent 2 are. Do they correspond to arbitrary labels or the turns that the agent takes in the game?
>
> We have clarified that Agent 1 is the agent who consistently goes first in negotiation.
>
> >Why is Agent 1 the one who triumphs in the no-communication channel game? Is there any advantage to going first generally?
>
> Actually, Agent 2 typically triumphs in the no-communication setup. We do not believe there is any significant advantages to either going first or second, as demonstrated by the fact that self-interested agents seem to be able to negotiate fairly in this environment.
>
> >Where are the tests of robustness on the curves demonstrated in Figure 2a?
>
> We have re-run the experiments with 20 different random seeds, and have updated Figure 2a to show their averaged results. The uncertainty estimates are unfortunately only visible in the linguistic communication setup, as the other communication protocols seem to give rise to very stable training.
>
> >Has figure 2b been cherry picked? This should be demonstrated over many different negotiations with error bars.
>
> Figure 2b shows the average reward per turn over 1000 different negotiations. We present results from training 20 seeds (1280 test negotiations per seed). We have added interquartile ranges at every timestep, and clarified that they are averaged results in the text.
>
> >Protocols could be designed that would allow the agents to communicate their utilities over this channel (within 4 turns), so the fact they don't suggests it is the learning procedure that is not able to find this optimum.
>
> We agree. However, discovering the optimal protocol through self-play RL is a significantly harder problem than designing one with knowledge of the optimal structure. For example, the pre-grounded nature of the proposal channel, combined with its lower information bandwidth, means that random exploration is less likely to find the optimal communication protocol.
>
> >Why are the 10 turn games even included in this table? It seems that this was dismissed in the environment setup section, due to the first mover advantage.
>
> We include the 10 turn results so that we can include the key self-interested and prosocial results in the same table. We also wanted to demonstrate how strong the first mover advantage was. We have moved these results to the appendix however in the new draft.
> We have also changed both tables to return mean and interquartile range as suggested.
>
> >Why do self-interested agents for 10 turns on the linguistic channel terminate early?
> We believe that when enough information is exchanged, the agents make effective proposals and thus do not need to negotiate further.
>
> >Both Figure 4 and Table 3 use Agent 1 and Agent 2 rather than Agent A and Agent B and is not clear whether this is a mistake or Agent 1 is different from Agent A.
>
> We have corrected this.
>
> >The no-communication baseline is referred to in the text but the results are not shown in the table.
>
> We have added the no-communication baseline figure to the text.
>
> >There are no estimates of the uncertainty of the results in table 3, how robust are these results to different initial conditions?
>
> These results are averaged across 10 batches of 128 games in each batch. We have clarified this, and added standard deviations to the table.

---

### Public Comment · ~Hugh_Nicholas_Perkins1 · 2017-10-30
**Misc comments 1**

Nice paper :) So I'm trying to reproduce it :)

Some very detailed questions, for trying to reproduce it:
- in figure 2, what does 'x 10' mean? does it mean eg '10000' means '100000' steps?
  - opinion: maybe putting eg 'x 100,000' or 'thousands', could be clearer, easier to read, remove some zeros from the graph?
  - is a 'step' an episode (each episode comprising multiple turns/timesteps?), or is a step a timestep?
- training curves for prosocial agents? how long does it take to learn to use the communications channel?
- you refer to the 'hidden state of the lstm'. Does this include the cell? just the hidden state? If the latter, does this mean that the cell is initialized to zero at the start of each generated utterance? If the former, how are you handling the associated doubling of embedding size?
- you say the vocabulary has 10 tokens. Does this include a termination token? or there are 11 possible tokens, 10 of which are part of the vocab, and one of which is termination?
   - similarly, does the utterance length 6 include the termination token, or is the length 6 + termination token?

---

> ### Author Response · Authors · 2017-10-30
> **Reply to misc comments 1**
>
> Hi Hugh
>
> Glad you liked the paper. Here's the answers:
>
> 1) Yeah, it means that the number should be multiplied by 10 to get the number of training steps.
> 2) We'll definitely consider this when revisions come around.
> 3) Each step is a training round, consisting of a batch of 128 games.
> 4) We didn't include training curves for prosocial agents - from memory, it took around 200k rounds to make use of the linguistic channel fully, and even more to use the proposal channel.
> 5) This is just the hidden state - the cell state is initialised to zero.
> 6) There are no termination tokens at all. There are 11 possible tokens, and each utterance is exactly of length 6.
>
> Hope this helps the replication.

---

> > ### Public Comment · ~Hugh_Nicholas_Perkins1 · 2017-10-31
> > **Thanks!**
> >
> > Thanks! :)

---

> > > ### Public Comment · ~Hugh_Nicholas_Perkins1 · 2017-11-01
> > > **Cite Entropy Regularization reference?**
> > >
> > > Opinion: might be worth including the reference for Entropy Regularization? I think it is 'Asynchronous methods for deep reinforcement learning', perhaps? (this itself cites Williams and Peng, but I think the Mnih et al reference is likely more useable?).

---

> > ### Public Comment · ~Hugh_Nicholas_Perkins1 · 2017-11-03
> > **Replication repo so far**
> >
> > Just for info, replication so far: https://github.com/ASAPPinc/emergent_comms_negotiation/blob/master/ecn.py
> >
> > Experiment log: https://github.com/ASAPPinc/emergent_comms_negotiation/blob/master/explog.md
> >
> > (stuck in a local minimum for now. digging a bit...)

---

### Public Comment · ~Hugh_Nicholas_Perkins1 · 2017-11-05
**Misc questions 2**

Hi,

Training is gradually progressing, though I'm possibly missing some tricks currently.  Some questions on reproducibility:
- hold out set of 5 batches of 128 test games:
   - presumably this is disjoint from the training set?
   - what is then the algorithm for generating the training set, to ensure disjointness?
- given that training seems to plateau for extended periods of time, eg see https://github.com/ASAPPinc/emergent_comms_negotiation/blob/master/images/20171104_192343gpu2_proposal_social_comms.png?raw=true up to first 100 thousand episodes, how did you decide when training was finished?
- you report the fraction of reward obtained in table 1.  Some questions on this:
  - you report the standard deviation across samples in a single batch of 128 games. To what extent did the fraction of reward vary across entire training runs?
  - you state that for prosocial, proposal channel only, the reward fraction was 0.92. However, in my own training run, the reward plateau's at ~0.81, after ~50,000 episodes of 128 games, and stays at ~0.81 for the remaining ~800,000 episodes, https://github.com/ASAPPinc/emergent_comms_negotiation/raw/master/images/20171104_144936_proposal_social_nocomms_b.png?raw=true  Thoughts?  What things should I consider checking in order to improve on this?

Some other questions, not strictly necessary for reproducibility:
- how were the entropy regularization weight hyperparameters determined?
   - using gridsearch?
   - what was the search space of the grid search?
   - was grid search run for both prosocial and not prosocial? or just for one particular set of settings?
- It looks to me like the main thesis of the paper is the behavior of selfish vs pro-social agents, is this an approximately fair impression? For example, the paper states 'self-interested agents cannot ground cheap talk. When using the linguistic channel, agents do not negotiate optimally. Instead, the agents randomly alternate taking all the items, which is borne out by the oscillations in Figure 2a.' I wonder whether it might be useful to have similar curves for the prosocial agents, so we can easily see how the behavior of prosocial and not prosocial agents compares?
- why have length 6 for the comms channel? Presumably it mostly just needs to be able to represent the hidden utility, which is sufficient for the other agent to have complete world state information? (Edit: oh, I guess the additional 3 length is necessary in the 'no proposal' case, so that the agent can communicate its proposal?)
- disparity between number of symbols in utterance vocab (10), and the possible utilities (11)? (in an earlier reply you say there are 11 utterance tokens in fact?)

---

> ### Author Response · Authors · 2017-11-13
> **Reply**
>
> Hi Hugh
> >- hold out set of 5 batches of 128 test games:
> >  - presumably this is disjoint from the training set?
> >  - what is then the algorithm for generating the training set, to ensure disjointness?
>
> We generated the test games by fixing the seed of the RNG, generating 5 batches from our game environment generator, and then discarding any batches of games after this which overlapped with any of the test games.
>
> >- given that training seems to plateau for extended periods of time, eg see https://github.com/ASAPPinc/emergent_comms_negotiation/blob/master/images/20171104_192343gpu2_proposal_social_comms.png?raw=true up to first 100 thousand episodes, how did you decide when training was finished?
>
> We basically just extended training until we saw no further improvements.
>
> >- you report the standard deviation across samples in a single batch of 128 games. To what extent did the fraction of reward vary across entire training runs?
>
> Our results were stable across training runs - we did extensive prototyping in a development environment and swept across random seeds, and the results were identical. We then moved onto the environment presented in the paper, and the results carried over.
>
> >Thoughts?  What things should I consider checking in order to improve on this?
>
> The hyperparameter we found with the biggest impact on the final outcome was the strength of the entropy regularization, as this controls the tradeoff between exploration and exploitation: too low and our agents didn't explore enough to find the optimal negotiation policy, too high and our agents just didn't learn anything. We found a useful debugging technique to be to monitor the proportion of actions taken equal to the argmax action, to ensure that the agents are still exploring during periods when the reward has plateaued. We also change the test time policy from the training time policy - during test time, we just deterministically take the action with the highest probability.
>
> >- how were the entropy regularization weight hyperparameters determined?
> >   - using gridsearch?
> >   - what was the search space of the grid search?
> >   - was grid search run for both prosocial and not prosocial? or just for one particular set of settings?
>
> We found the optimal hyperparameter values with grid search, using values in {1e-4, 5e-4, 1e-3, 5e-3, 1e-2, 5e-2} for the termination and proposal policies, and {1e-5, 5e-5, 1e-4, 5e-4, 1e-3, 5e-3} for the utterance policy. We ran this for all agent reward schemes, and the same hyperparameter values worked regardless of the agent reward scheme.
>
> > I wonder whether it might be useful to have similar curves for the prosocial agents, so we can easily see how the behavior of prosocial and not prosocial agents compares?
>
> We have the training graphs for prosocial agents, but we did not include them in this revision due to space constraints. They mainly show the joint reward going fairly smoothly upwards as training goes on, and then plateauing at the end. We can add them to the next revision if necessary.
>
> > - disparity between number of symbols in utterance vocab (10), and the possible utilities (11)? (in an earlier reply you say there are 11 utterance tokens in fact?)
>
> The item utilities range between 0 and 5 inclusive, not 0-10. The symbols the agents can communicate with range between 0-10. We also ran an experiment on how the bandwidth of the utterance channel affected prosocial agent negotiation success, and essentially i- disparity between number of symbols in utterance vocab (10), and the possible utilities (11)? (in an earlier reply you say there are 11 utterance tokens in fact?)t seems that as long as there is plenty of spare capacity in the utterance channel then the agents can learn to negotiate fine.
>
> Hope this helps
> The authors

---

### Author Response · Authors · 2018-01-03
**Reply to all reviewers**

Thank you to all reviewers for their thoughtful feedback. We particularly thank the reviewers for agreeing with us that the negotiation task is an interesting task to study emergent communication. We would also like to thank the reviewers for their compliments regarding the clarity of the writing - we hope our responses carry on this theme.

We have uploaded a revised version of the manuscript to address the concerns raised by the reviewers. The most significant changes are:
  - We have added interquartile ranges to Table 1, and split out the 10 turn results to the appendix. We also carried out additional experiments with 20 different initial seeds for the agent weights and added this to Table 1.
  - We have added uncertainty estimates to Table 3
  - We added uncertainty estimates to Figures 2a and 2b.
  - We added transcripts of the negotiation setup for various setups to the main body of the paper, as the new Tables 2 and 3.

---

### Decision · Program_Chairs · 2018-01-29
**ICLR 2018 Conference Acceptance Decision**

**Decision:**

Accept (Poster)

**Comment:**

All reviewers agree the paper proposes an interesting setup and the main finding that "prosocial agents are able to learn to ground symbols using RL, but self-interested agents are not" progresses work in this area. R3 asked a number of detail-oriented questions and while they did not update their review based on the author response, I am satisfied by the answers.